# The Integrated Stress Response Is Tumorigenic and Constitutes a Therapeutic Liability in Somatotroph Adenomas

**DOI:** 10.3390/ijms232113067

**Published:** 2022-10-28

**Authors:** Zhenye Li, Yiyuan Chen, Xiaohui Yao, Qian Liu, Haibo Zhu, Yazhuo Zhang, Jie Feng, Hua Gao

**Affiliations:** 1Beijing Tiantan Hospital, Capital Medical University, Beijing 100070, China; 2Beijing Neurosurgical Institute, Capital Medical University, Beijing 100070, China; 3Shanxi Provincial People’s Hospital, Taiyuan 030000, China; 4China National Clinical Research Center for Neurological Diseases, Beijing 100070, China; 5Key Laboratory of Central Nervous System Injury Research, Beijing 100070, China

**Keywords:** somatotroph adenomas, integrated stress response, tumorigenesis, EIF2

## Abstract

Somatotroph adenomas are the leading cause of acromegaly, with the nearly sparsely granulated somatotroph subtype belonging to high-risk adenomas, and they are less responsive to medical treatment. The integrated stress response (ISR) is an essential stress-support pathway increasingly recognized as a determinant of tumorigenesis. In this study, we identified the characteristic profiling of the integrated stress response in translocation and translation initiation factor activity in somatotroph adenomas, normal pituitary, or other adenoma subtypes through proteomics. Immunohistochemistry exhibited the differential significance and the priority of eukaryotic translation initiation factor 2β (EIF2β) in somatotroph adenomas compared with gonadotroph and corticotroph adenomas. Differentially expressed genes based on the level of EIF2β in somatotroph adenomas were revealed. MetaSape pathways showed that EIF2β was involved in regulating growth and cell activation, immune system, and extracellular matrix organization processes. The correlation analysis showed Spearman correlation coefficients of r = 0.611 (*p* < 0.001) for EIF2β and eukaryotic translation initiation factor 2 alpha kinase 1 (HRI), r = 0.765 (*p* < 0.001) for eukaryotic translation initiation factor 2 alpha kinase 2 (PKR), r = 0.813 (*p* < 0.001) for eukaryotic translation initiation factor 2 alpha kinase 3 (PERK), r = 0.728 (*p* < 0.001) for GCN2, and r = 0.732 (*p* < 0.001) for signal transducer and activator of transcription 3 (STAT3). Furthermore, the invasive potential in patients with a high EIF2β was greater than that in patients with a low EIF2β (7/10 vs. 4/18, *p* = 0.038), with a lower immune-cell infiltration probability (*p* < 0.05). The ESTIMATE algorithm showed that the levels of activation of the EIF2 pathway were negatively correlated with the immune score in somatotroph adenomas (*p* < 0.001). In in vitro experiments, the knockdown of EIF2β changed the phenotype of somatotroph adenomas, including cell proliferation, migration, and the secretion ability of growth hormone/insulin-like growth factor-1. In this study, we demonstrate that the ISR is pivotal in somatotroph adenomas and provide a rationale for implementing ISR-based regimens in future treatment strategies.

## 1. Introduction

Somatotroph adenomas are a kind of functional pituitary adenomas that cause acromegaly due to the excessive secretion of growth hormone (GH) and insulin-like growth factor (IGF-1) [1]. Excess GH/IGF-1 in somatotroph adenoma patients causes the overgrowth of some tissues and multiple metabolic abnormalities by affecting body composition, protein dynamics, and molecular mechanisms in adults [2]. The mortality risk is related to the duration of the illness before diagnosis and the lesional degree of cardiovascular disease and diabetes at the time of diagnosis [3]. Improvements in diagnostic modalities have increased the prevalence of somatotroph adenomas through the identification of active acromegaly (high IGF-1 and/or GH levels) in clinically asymptomatic patients, also called silent somatotroph adenomas [3]. Compared to other subtypes, up to 50% of somatotroph adenoma patients have clear phenotypic changes, whereas another ~50% have either mild symptoms or are visibly asymptomatic for metabolic abnormality, including polyamine anabolism and sugar anabolism [4]. Initial therapy generally constitutes transsphenoidal surgery, although the remission rate is approximately 50%; somatostatin analog (SSA) therapy is reserved for cases not cured by surgery [5]. The WHO 2017 classification suggests that subtypes of pituitary adenoma, such as male lactotrophs, silent corticotroph and Crooke cells, sparsely granulated somatotrophs, and silent plurihormonal POU Class 1 Homeobox 1-positive tumors, should be considered as “high risk” tumors [6].

The integrated stress response (ISR) is an essential stress-support pathway increasingly recognized as a determinant of tumorigenesis via tuning protein synthesis rates [7]. The various stresses are sensed by four specialized kinases—eukaryotic translation initiation factor 2 alpha kinase 1 (HRI), eukaryotic translation initiation factor 2 alpha kinase 2 (PKR), eukaryotic translation initiation factor 2 alpha kinase 3 (PERK), and eukaryotic translation initiation factor 2 alpha kinase 4 (GCN2)—that converge on the phosphorylation of a single serine on the eukaryotic translation initiation factor eIF2 [8]. The salient feature of ISR signaling lies in the modulation of the cellular concentration of the ternary complex (TC), which is composed of the heterotrimeric eukaryotic translation initiation factor and eukaryotic translation initiation factor 2 (EIF2), consisting of an α, β, and γ subunit in equal molar amounts [9]. Phosphorylation of the eIF2 subunit regulates the heterotrimer activity [10]. Oncogenic roles of eIF2α are well-known in chronic myeloid leukemia [11], colorectal cancer [12], prostate cancer [13], and so on. eIF2β forms part of the nucleotide-binding pocket in eukaryotes and appears to bind GTP or GDP [14]. Integrated genomic data in breast cancer has shown that eIF2β is highly amplified and correlates to the aggressive behavior of luminal breast tumors [15]. eIF2γ has a catalytic site for GTPase activity and is involved in binding of the ternary complex to mRNA; the eIF2γ variant is associated with hypothalamo-pituitary development, hypopituitarism, and pancreatic dysfunction and glucose regulation [16]. The ISR is a double-edged sword with both pro-survival and pro-death activities, and regulation of ISR signaling components is a promising strategy in cancer therapy [17,18].

While the ISR is upregulated in many cancers, its precise function in tissue-specific cancer initiation and progression remains poorly understood [19]. In this study, we explored somatotroph adenomas using multi-omic methods and showed the unique characteristics of the eIF2 pathway and ribosome-related genes in somatotroph adenoma patients compared to patients with other subtypes of pituitary adenomas. We demonstrated the relationship of the ISR with disease phenotype and immune cell infiltration in somatotroph adenomas together with the strong therapeutic benefits of its pharmacological inhibition.

## 2. Results

### 2.1. Clinical Characteristics of Patients

Our study included 144 patients: 33 somatotroph adenomas, 75 gonadotroph adenomas, and 36 corticotroph adenomas (Table 1). There were 79 males and 65 females, with an average age of 49.1 ± 1.01 years (range: 17–69). The average tumor size was 11.24 ± 1.53 cm^3^ (range: 0.74–42.76), including 2 microadenomas, 124 macroadenomas, and 18 giant adenomas. According to Knosp staging, there were 67 invasive cases and 77 non-invasive cases. The average follow-up time was 56.6 ± 10.7 months (range: 42–78), with five-year recurrence rates as follows: 7/33 (21.2%) for somatotroph adenomas, 10/75 (13.3%) for gonadotroph adenomas, and 13/36 (36.1%) for corticotroph adenomas. There were more female patients and a greater invasion and recurrence chance in corticotroph adenoma and more younger patients with somatotroph adenoma (*p* < 0.05).

In this cohort, the gender balance was 8:28 in CORT, 55:20 in GONA, and 16:17 in SOMA (*p* < 0.001). That is, there were more female patients in CORT and more male patients in GONA. The average age was 50.75 ± 2.10 in CORT, 52.85 ± 1.28 in GONA, and 43.58 ± 1.89 in SOMA (*p* < 0.001). The youngest patients were in the SOMA group, and there was no statistical difference in age between CORT and GONA. The recurrence rate was 13/36 (36.1%) in CORT, 10/75 (13.3%) in GONA, and 47/33 (21.2%) in SOMA (*p* < 0.001). The worst prognosis was in CORT compared with GONA and SOMA.

### 2.2. Characteristic Profiling in Somatotroph Adenomas Compared to Normal Pituitary, Corticotroph Adenomas, and Gonadotroph Adenomas

Differentially expressed proteins were identified by nanoLC-MS/MS among five normal pituitary tissues, five somatotroph adenomas, five gonadotroph adenomas, and five corticotroph samples. Proteomics analysis identified 46,383 peptides that mapped to 4751 proteins. A total of 274 proteins were differentially expressed (*p* < 0.05, with an iTRAQ ratio >1.5 or <0.7); 130 were upregulated; and 144 were downregulated in somatotroph adenomas, as shown in Figure 1A,B. Using the Metascape database, pathway and process enrichment analyses showed that the top three pathways were SRP-dependent cotranslational protein targeting to the membrane, peptide chain elongation, and eukaryotic translation elongation, as shown in Figure 1C. The top three GO biological processes were cytoplasmic translation, protein localization to the plasma membrane, and regulation of neurotransmitter levels, as shown in Figure 1D. The summary of enrichment analyses in TRRUST showed that the differentially expressed proteins were regulated by STAT3, REST, TFAP2A, SNAI1, etc., as depicted in Figure 1E.

Somatotroph adenoma and lactotroph both belong to the PIT-1 lineages of PAs. In this study, we analyzed the difference between somatotroph and macro-lactotroph adenomas, as shown in Appendix A. The volcano map shows the top 20 DEGs in Appendix A. Using the Metascape database, DEG pathway enrichment shows the top 10 GO biological processes in Appendix A and the top 10 KEEG pathways in Appendix A.

EIF2 is a key component of the ternary complex whose role is to deliver initiator tRNA into the ribosome. Mass-spectrometric data showed increased subunit α/β and reduction in γ subunit, mainly subunit β. Immunohistochemistry exhibited the differential significance and the priority of EIF2β in somatotroph adenomas compared with gonadotroph and corticotroph adenomas, as depicted in Figure 2.

Correlation analysis showed that the correlation coefficient of EIF2β and STAT3 was 0.541 (Appendix A) (*p* < 0.001).

### 2.3. Differentially Expressed Gene-Enrichment Analysis between EIF2β Groups

According to the H-score of EIF2β (H-score: 114), patients with somatotroph adenomas were divided into a high-EIF2β group (*n* = 10) and a low-EIF2β group (*n* = 18); six samples were excluded because of staining loss. Based on different levels of EIF2β, there were 244 differentially expressed genes (Ι fold changeΙ ≥ 1, *p* < 0.05). The top three pathways were cytokine signaling in the immune system, hemostasis, and extracellular matrix organization (Figure 3A). The top three GO biological processes were complement activation, classical pathway complement activation, and humoral immune response mediated by circulating immunoglobulins (Figure 3B). Correlation analysis showed the following Spearman correlation coefficients: EIF2β and HRI (r = 0.611, *p* < 0.001), PKR (r = 0.765, *p* < 0.001), PERK (r = 0.813, *p* < 0.001), GCN2 (r = 0.728, *p* < 0.001), and STAT3 (r = 0.732, *p* < 0.001) (Figure 3C).

Furthermore, we analyzed the clinical relevance of EIF2β in 28 patients with somatotroph adenoma (Table 2). Patients with a high EIF2β had a greater invasive potential (*p* = 0.038). There was a tendency toward a more sparsely granulated adenoma phenotype in 28 somatotroph adenoma patients with a high EIF2β than in patients with a low EIF2β.

### 2.4. Immunologic Signatures in Patients with Somatotroph Adenomas

First, we described the immune cell types and percentages with a heatmap for patients with somatotroph adenomas (Figure 4A). The associations between the infiltrate levels of immune cell signatures and levels of EIF2β are illustrated in Figure 4B; there was a statistically significant difference in lymphocytes, macrophages, and dendritic cells between patients with a high EIF2β and patients with a low EIF2β (*p* < 0.05) (Figure 4C). Moreover, according to the unpaired t test, there were significant differences in the immune score (*p* = 0.020) and ESTIMATE score (*p* = 0.024) between patients with a high EIF2β and patients with a low EIF2β (Figure 4D).

### 2.5. Enumeration and Spatial Distribution of Immune Cells in Pituitary Adenomas Determined by Immunohistochemistry Staining

Innate immune cells and adaptive immune cells are very important for host defense, and the balance of immune responses is critical in controlling cancer progression. An immunohistochemistry experiment revealed that the mean number of immune cells was 7.41 ± 3.87% (range 0–32.9%) in somatotroph adenomas, 5.35 ± 3.65% (range 0–27.1%) in gonadotroph adenomas, and 4.97 ± 3.27% (0–42.9%) in corticotroph adenomas (Figure 5).

The immune cells were mainly distributed in the tumor parenchyma, followed by the tumor stroma and perivascular region. CD8+ cells were mainly distributed in the tumor stroma (69/144, 47.9%), followed by the tumor parenchyma (62/144, 43.1%) and perivascular area (13/144, 9.0%). CD20+ cells were mainly distributed in the tumor stroma (103/144, 71.5%), tumor parenchyma (21/144, 14.6%), and perivascular region (20/144, 13.9%). CD163+ cells were distributed in the tumor parenchyma (68/144, 47.2%), tumor stroma (58/144, 40.3%), and perivascular area (18/144, 12.5%). In this study, the cutoff value for infiltration was taken as ≥5% positive cells. There was a higher immune-cell infiltration probability (24/69 vs. 13/75, *p* = 0.017) in functional pituitary adenomas than in non-functional pituitary adenomas.

### 2.6. Effects of RNAi-EIF2β on Cell Proliferation and Growth Hormone Release by GH3 Cells

To test the role of EIF2β in somatotroph adenomas, we constructed the RNA interference fragments of EIF2β and packaged the viral vectors. The interfering efficiency of shRNA1–4 were 0.97 ± 0.15, 0.57 ± 0.02, 0.47 ± 0.03, and 0.91 ± 0.19-fold, respectively, of control GH3 cells after 72 h transient transfection (Figure 6A) (*p* < 0.05). Western blot experiments also showed the same tendency (Figure 6B). The sh2 and sh3 fragments were used in further experiments. The cell viabilities in RNAi groups were 78.4 ± 4.7% and 84.5 ± 5.1% of that in the control group after 24 h infection, 67.3 ± 3.9% and 71.5 ± 4.1% after 48 h infection, and 60.4 ± 3.8% and 65.2 ± 3.7% after 72 h infection, respectively (Figure 6C, *p* < 0.05). ELISA showed that GH levels in the cell culture were 5.7 ± 1.1 ng/mL and 6.1 ± 0.9 ng/mL in the RNAi-EIF2β groups compared with 11.7 ± 0.7 ng/mL in the control GH3 group (*p* < 0.05) after 72 h treatment (Figure 6D). IGF-1 levels in the cell culture were 7.7 ± 1.2 ng/mL and 9.2 ± 1.1 ng/mL in the RNAi-EIF2β groups compared with 16.3 ± 0.9 ng/mL in the control group (*p* < 0.05) after 72 h treatment (Figure 6E). Transwell experiments showed that the trans-membrane-positive cells in the RNAi-sh2 group (226 ± 27/high field) and RNAi-sh3 group (254 ± 32/high field) were reduced compared with the control group (403 ± 67/high field) (Figure 6F). We also found a decline in STAT3 and STAT5b in the sh2 and sh3 groups (Figure 6G).

## 3. Discussion

Somatotroph adenomas usually induce a decrease in body fat, osteoporosis, and increase water retention. Recent studies suggest that abnormal fat metabolism promotes protein anabolism and relieves amino acid oxidation. Protein synthesis dysregulation is a burgeoning field in cancer biology. The ISR is an essential stress-support pathway, increasingly recognized as a determinant of tumorigenesis by fine-tuning protein-synthesis rates. Translation initiation lies downstream to deregulated signal pathways in cancer and is regulated by activated oncogenes or mutated tumor suppressors. Therefore, controlling protein synthesis has significant potential for developing innovative therapy in somatotroph adenomas.

In this study, we found significant differences in the eIF2 pathway in somatotroph adenomas compared to normal pituitary and other subtypes of pituitary adenomas through multi-omics analysis. Furthermore, we evaluated the effects of the eIF2 pathway on cell proliferation, invasion, and growth hormone release through ex vivo experiments.

Our understanding of the central mechanisms that govern the ISR has advanced vastly. The ISR’s central regulatory hub lies in the EIF2-EIF2B complex, which controls the formation of the eEIF2/GTP/methionyl-initiator tRNA TC, a prerequisite for initiating new protein synthesis [7]. EIF2 and EIF2B are complex proteins of three (α–γ) and five (α–ε) nonidentical subunits, respectively [20]. Through integrated multi-omics analysis, we found an abnormally increased profile of ribosomal proteins in somatotroph adenomas compared with normal pituitary glands or other subtypes of pituitary adenomas. In the diversity of differentially expressed proteins, the top three pathways were: SRP-dependent cotranslational protein targeting the membrane, peptide chain elongation, and eukaryotic translation elongation by KEGG pathway enrichment, as shown in Figure 1C. Moreover, we found an imbalance in the EIF2 complex in somatotroph adenomas, i.e., increased α/β subunit and reduction in the γ subunit of eIF2.

Oncogenic roles of eIF2ɑ have been extensively studied in several human cancers [21]; however, few studies demonstrate the roles of eIF2β in carcinogenesis. An integrated genomic search of novel therapeutic targets for breast cancer indicated that eIF2β is highly amplified in luminal breast tumors, which are an aggressive subtype of breast tumors [22]. In in vitro experiments, the knockdown of eIF2β manipulations induced G1 cell cycle arrest in both NCI-H358 and NCI-H460 cells [23]. Deleting the polylysine stretches in the amino-terminal region of eIF2β decreased protein synthesis, inhibited cell proliferation and viability, increased cell death, and induced G2 cell cycle arrest in Hek293TetR cells [24]. Interestingly, we noticed that the levels of EIF2β in somatotroph adenomas were related to the disease phenotype. Patients with a high EIF2β had a higher invasive potential and greater chances of sparsely granulated tumors than patients with low immune-cell infiltration with low EIF2β. The phosphorylation of eIF2α is carried out by a family of four kinases, PERK, PKR, GCN2, and HRI, which primarily respond to a distinct type of stress or stresses [25]. In this study, we observed a high correlation between EIF2ɑ/β and PKR and PERK expression, suggesting that simultaneous upregulation of these molecules increases translation initiation, contributing to enhanced synthesis and secretion of GH/IFF-1 by somatotroph tumor cells. We found a decrease in cell viability and the level of GH and IGF-1 in the cell culture supernatant of GH3 cells in both sh2-EIF2β and sh3-EIF2β groups.

Enrichment analysis in TRRUST showed that the differentially expressed proteins were regulated by STAT3, REST, TFAP2A, SNAI1, etc. It was reported that STAT3 specifically binds the GH promoter and induces transcription in the GH3 cell line [26]. Additionally, C188-9, an inhibitor of STAT3, could downregulate the levels of eIF2 and STAT3 in GH3 cell lines. We confirmed a positive correlation between EIF2β and STAT3 based on transcriptome data and immunohistochemistry staining. In vitro experiments verified that RNAi-EIF2β relieved cell proliferation, invasion, and secretion of GH/IGF-1. Western blot experiments also verified the relationship between EIF2β and STAT3. We hypothesized that EIF2β was mainly involved in tumor-related pathways in somatotroph adenomas, including the cell adhesion molecule pathway and the JAK-STAT signaling pathway. We speculated that eIF2β might inhibit T cell-mediated immunity by promoting the Treg response. In addition, the expression of eIF2β was positively correlated with STAT3 and STAT5b. Related studies have shown that some signaling molecules are involved in the M2 polarization of macrophages, such as PI3K/AKT-ERK signaling, STAT3, HIF1α, and STAT6. These results suggested that eIF2β might regulate tumor macrophage infiltration, affecting the tumor microenvironment.

The tumor immune microenvironment is increasingly being recognized as a key contributor to tumorigenesis and tumor progression and recurrence. Cancer cells, stromal cells, and immune cells, along with their extracellular factors, have profound effects on either promoting or repressing anti-cancer immunity. These effects were mostly detected by immunohistochemistry; immune cell infiltration varied greatly among samples, which indicated that pituitary-hormone-secreting adenomas have substantial heterogeneity. For example, it has been reported that CD68+ macrophage infiltration was higher than that in corticotroph adenomas and densely granulated somatotroph adenomas [27]. In this study, we found a greater probability of immune cell infiltration in corticotroph adenomas and somatotroph adenomas compared with gonadotroph adenomas. Since corticotroph adenomas and somatotroph adenomas have greater invasive potential, we speculated that immune cell infiltration was related to invasive behavior. However, we found a negative relationship between the levels of EIF2β and immune score in 28 patients with somatotroph adenoma based on the “ESTIMATE” algorithm, which was further validated by immunohistochemistry data. The enriched GO terms by DEG analysis for biological processes were complement activation, classical pathway complement activation, and humoral immune response mediated by circulating immunoglobulins, whereas KEGG pathway terms enriched by DEG analysis were cytokine signaling in the immune system, hemostasis, and extracellular matrix organization; these may all have roles in the modulation of inflammatory infiltration in tumor tissue. The specific mechanism by which inflammatory infiltration leads to tumor invasion requires further investigation; this may also be relevant for further research into and development of targeted drugs for somatotroph adenomas.

However, our study has some limitations. First, the small sample size may have contributed to a bias error; the relationship between eIF2β and sparse granularity needs further confirmation in large samples. Moreover, the transcriptome data analyzed were generated through bulk RNA sequencing; bulk sequencing is likely to cause loss of some information because of the heterogeneous nature of tumor tissues, and this may be explored in more detail with single-cell RNA-seq techniques.

In conclusion, in the present study, we uncovered the unique characteristics of the eIF2 complex and abnormal integrated stress response through omics data in somatotroph adenoma patients. Furthermore, we characterized the landscape of the somatotroph adenoma microenvironment using different methodologies, including immunohistochemistry and transcriptomics. We speculated that eIF2β promoted the synthesis and secretion of GH/IGF-1 in somatotroph tumor cells and inhibited immune cell infiltration.

## 4. Material and methods

### 4.1. Patient Sample Acquisition and Cell Lines

All tumor samples were obtained following transsphenoidal surgery performed at Beijing Tiantan Hospital, Beijing, China, from May 2015 to December 2020. Fresh tumor samples were stored in liquid nitrogen. A total of 144 patient samples (age range: 17–69 years) were diagnosed as pituitary adenomas, including 33 somatotroph adenomas, 75 gonadotroph adenomas, and 36 corticotroph adenomas. The pituitary adenoma types were diagnosed according to the 2017 World Health Organization classification of tumors of endocrine organs, including pituitary transcription factors and hormone content, by immunohistochemistry staining. Somatotroph adenomas are divided into two distinct and clinically relevant histologic subtypes, densely and sparsely granulated, based on the density of GH-containing secretory granules and the presence of fibrous bodies. Five normal pituitary glands were obtained from a donation program. Protocols were approved by the Internal Review Board of Beijing Tiantan Hospital affiliated with Capital Medical University, and the study was conducted in accordance with the principles laid down in the Declaration of Helsinki (no. KY2016-035-01).

GH3 cells were purchased from ATCC and cultured in F-12K medium (ATCC, Manassas, VA, USA) with 2.5% fetal bovine serum and 10% horse serum in a humidified incubator at 37 °C and 5% CO_2_.

### 4.2. Protein Preparation and Proteomics Analysis

Protein preparation and proteomics analysis were completed in accordance with the methods used in our previous study [28]. A mass of 100 μg of each pooled sample was then denatured, reduced, and alkylated as described in the iTRAQ protocol (Thermo Fisher, Waltham, MA, USA) and digested overnight with 0.1 µg/µL trypsin solution at 37 °C. Peptides were analyzed by nanoLC-MS/MS on a Q Exactive mass spectrometer (Thermo Fisher, USA). Chromatography was performed with solvent A (Milli-Q (Millipore, Billerica, MA, USA) water with 2% acetonitrile and 0.1% formic acid) and solvent B (90% acetonitrile with 0.1% formic acid). The Q Exactive instrument was operated in information-dependent data acquisition mode to switch automatically between MS and MS/MS acquisition. MS spectra were acquired across the mass range of 350–2000 *m/z*. All MS/MS data were analyzed using Mascot (Matrix Science, London, UK; version 2.3.0). Carbamidomethylation of cysteine residues was specified as a fixed modification. Oxidation of methionine residues was specified in Mascot as a variable modification. Only proteins with a *p* < 0.05 were accepted.

### 4.3. RNA Microarray and PCR Verification

Microarray hybridization and qRT-PCR were conducted in accordance with the methods used in our previous study [28]. Total RNA of 30 samples was extracted and purified using mirVana™ miRNA Isolation Kit (Ambion, Austin, TX, USA) following the manufacturer’s instructions. Labeled cRNA was purified using an RNeasy mini kit (QIAGEN, Hilden, Germany). Each slide was hybridized with 1.65 μg Cy3-labeled cRNA using a Gene Expression Hybridization Kit (Agilent Technologies, Santa Clara, CA, USA) in a Hybridization Oven (Agilent Technologies), according to the manufacturer’s instructions. After 17 h hybridization, slides were washed in staining dishes (Thermo Shandon, USA) with a Gene Expression Wash Buffer Kit (Agilent Technologies), following the manufacturer’s instructions. Slides were scanned with an Agilent Microarray Scanner using default settings: Dye channel: green, scan resolution = 3 μm, PMT 100%, 20 bit. Data were extracted with Feature Extraction software 10.7 (Agilent Technologies). Raw data were normalized using the Quantile algorithm and limma package in R. The selected mRNAs were grouped into functional categories based on the Gene Ontology database (GO: http://www.geneontology.org/, accessed on 1 August 2022), and functional pathways (KEGG and BIOCARTA) were also analyzed using an online SAS analysis system.

qRT-PCR was performed on the Applied Bio-systems 7500 Fast System (Life Technologies, Carlsbad, CA, USA). The fold-change in differential expression for each gene was calculated using the comparative CT method (2^−∆∆CT^ method), R package: “pcr”, https://github.com/MahShaaban/pcr, accessed on 5 May 2020 reference gene: “*GAPDH*,” reference group: “Somatotroph Adenoma” [29].

### 4.4. Tissue Microarray Construction and Immunohistochemistry Staining

A total of 145 formalin-fixed, paraffin-embedded tissue blocks were sliced and stained with hematoxylin–eosin, and slides were prepared. Three core biopsies, 2.0 mm in diameter, were selected from the paraffin-embedded tissue. The cores were transferred to tissue microarrays using the Minicore from Mitogen (UK). The tissue microarrays were cut into 4 μm sections. Primary antibodies anti-eIF2β (1:300, Abcam, Cambridge, UK), anti-STAT3 (1:1000, Abcam), anti-CD4 (1:300, Abcam), anti-CD8 (1:400, Abcam), anti-CD20 (1:100, LEICA, Wetzlar, Germany), anti-CD86 (1:800, Abcam), and anti-CD163 (1:600, Abcam) were used. The H-score was obtained by multiplying the staining intensity with a constant to adjust the mean to the strongest staining section (H-score = 3 × (percentage of strong staining); 1.0 (% weak), 2.0 (% moderate), 3.0 (% strong)) to give a score ranging from 0 to 300.

### 4.5. SDS-PAGE and Western Blot Analyses

The 10 mg samples were lysed in lysate buffer containing 1% Nonidet *p*-40 (Calbiochem, Germany) with protease- and phosphatase-inhibitor cocktails (Roche) overnight at 4 °C. Total extracts were centrifuged for 30 min at 12,000× *g* and 4 °C, and protein concentration was determined using the BCA method (Pierce Biotechnology). An aliquot of 40 μg protein per lane was loaded onto 10% Bis-Tris SDS-PAGE gels, separated electrophoretically, and blotted onto polyvinylidene fluoride (PVDF) membranes. Different blots were incubated with antibodies against anti-eIF2β (1:1000, Abcam), STAT3 (1:2000, Abcam), STAT5b (1:2000, Abcam), and GAPDH (1:8000, Sigma, St. Louis, MI, USA), followed by secondary antibodies tagged with horseradish peroxidase (Santa Cruz Biotechnology, Dallas, TX, USA). Blots were visualized by enhanced chemiluminescence, and densitometry was performed with Amersham Imager 600. Analysis of GAPDH levels was used as a loading control.

### 4.6. Construction of Expression Plasmids and Virus Packaging

This analysis identified two stretches of 21 nucleotides in the eIF2β region as potential shRNA targeting sequences: shRNA-2: 5′-GGAGGACGACCTTGACATT-3′, shRNA-3: 5′-CCCAAACATCTCCTTGCAT-3′. Scrambled targeting sequence with no known mammalian homology, 5′-GGATTGATTCAACACGGAAGA-3′, was used as the negative control. DNA oligonucleotides were synthesized, annealed, and cloned into Plvx-shRNA2-ZsGreen1-T2A-Puro (Oligobio, Beijing, China). The inserts of the clones were verified by PCR and sequencing. Positive viruses for eIF2β shRNA and control shRNA expression were named sh2 and sh3, respectively. Plvx-shRNA2-T2A-puro- eIF2β vector supernatants were generated by co-transfecting 15 μg Plvx-shRNA2-T2a-puro shRNA vector, 11.25 μg of psPAX2 transfer vector, and 3.75 μg of Pmd2.G into HEK293T cells in a 12-well plate using linear PEI reagent. Supernatants were collected at 48 h after transfection and filtered through a 0.45 μm polyvinylidene fluoride (PVDF) syringe filter (Corning Life Sciences, Tewksbury, MA, USA). To determine the titer, we infected 3 × 10^4^ cells in a 48-well plate with serial dilutions of the Plvx-shRNA2-T2A-puro- eIF2β vector supernatants in the presence of polybrene (10 μL) (Sigma-Aldrich, St. Louis, MO, USA) and analyzed them 3 days post-transduction by flow cytometry analysis using the BD FACS Fortessa (BD Biosciences, San Jose, CA, USA). Transductions of HEK293T cell lines were performed on 2 × 10^4^ cells/well in a 24-well plate followed by flow cytometric analysis of Venus expression 3 days post-transduction.

### 4.7. Cell Proliferation and Migration Assays

GH3 cells were adjusted to a density of 1 × 10^5^ cells/mL. An aliquot of 100 μL of cell suspension was plated into each well of 96-well plates and cultured overnight. At 24, 48, and 72 h post-transfection, 20 μL 3-(4,5-diethylthiazol-2-yl)-5-(3-carboxymethoxyphenyl)-2-(4-sulfophenyl)-2H-etrazolium inner salt (MTS) solution was added to each well, followed by further incubation for 4 h. Absorbance of each well at 490 nm was measured using an ELISA plate reader (Thermo, USA). Cell migration was measured using fibronectin and Matrigel-coated polycarbonate filters, respectively, and modified transwell chambers (Corning, MA, USA). GH3 cells (5 × 10^4^ cells) were added into the upper chambers. After 24 h of incubation, migrating cells that adhered to the lower membrane were fixed in 4% paraformaldehyde and stained using hematoxylin (Zhongshan Company, Beijing, China). No invading cells were removed with cotton swabs. Images were photographed in three randomly selected fields of view at 200× magnification. Experiments were performed in triplicate.

### 4.8. ELISA

The levels of GH and IGF-1 were detected using an ELISA kit (APPLYGEN) according to the protocol. An aliquot of 10 μL of cell culture supernatant per well was used. The absorbance of each well was measured at 450 nm using an ELISA plate reader (Thermo Fisher). The amount of GH and IGF-1 was calculated from a standard curve prepared using the recombinant protein. Positive controls were supplied in the kit.

### 4.9. Statistical Analysis

Chi-square test and Fisher’s exact test were used to determine the significance of categorical variables. One-way ANOVA was applied to the immunohistochemical results in patients or in vitro experiments. All *p* values are two-sided, and 0.05 was applied as the significance level.

## Figures and Tables

**Figure 1 ijms-23-13067-f001:**
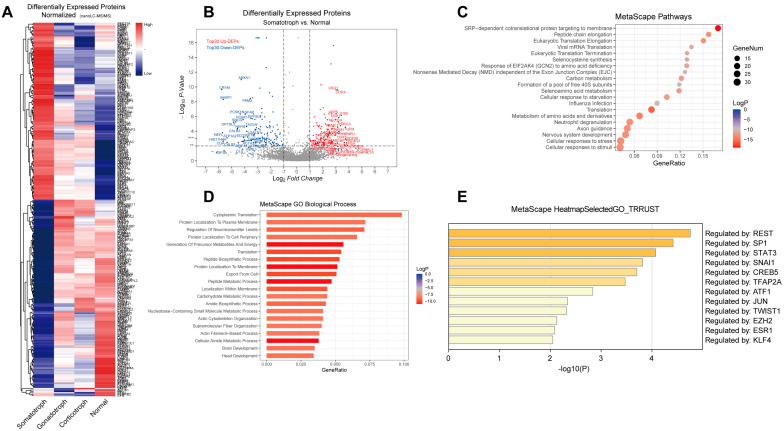
Proteomics analysis of normal pituitary gland and corticotroph adenomas, gonadotroph adenomas, and somatotroph adenomas. (**A**): A heatmap of differentially expressed proteins (FC > 1.5 or FC < 0.67 and FDR q < 0.1). High expression is shown in red; low expression is shown in blue. (**B**): Volcano map of differentially expressed proteins.** *p* < 0.01, *** *p* < 0.001 (**C**): KEGG pathway analysis based on differentially expressed proteins. (**D**): The most enriched GO terms. (**E**): Enrichment analysis in TRRUST showing differentially expressed proteins regulated by STAT3, REST, TFAP2A, and SNAI1 based on the Metascape database.

**Figure 2 ijms-23-13067-f002:**
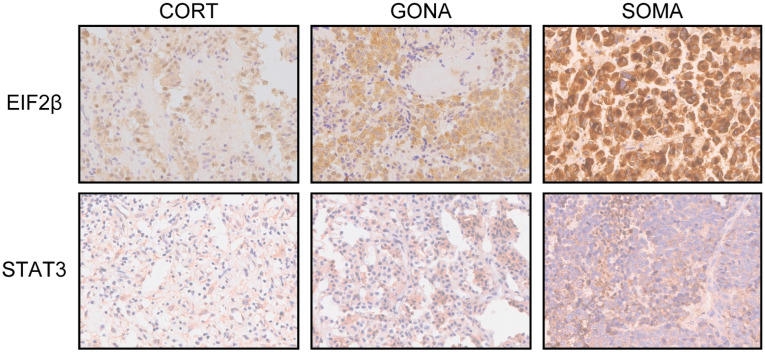
Immunohistochemistry images of EIF2β and STAT3 in corticotroph adenomas, gonadotroph adenomas, and somatotroph adenomas. CORT: corticotroph adenomas; GONA: gonadotroph adenomas; SOMA: somatotroph adenomas (×: 40 fold).

**Figure 3 ijms-23-13067-f003:**
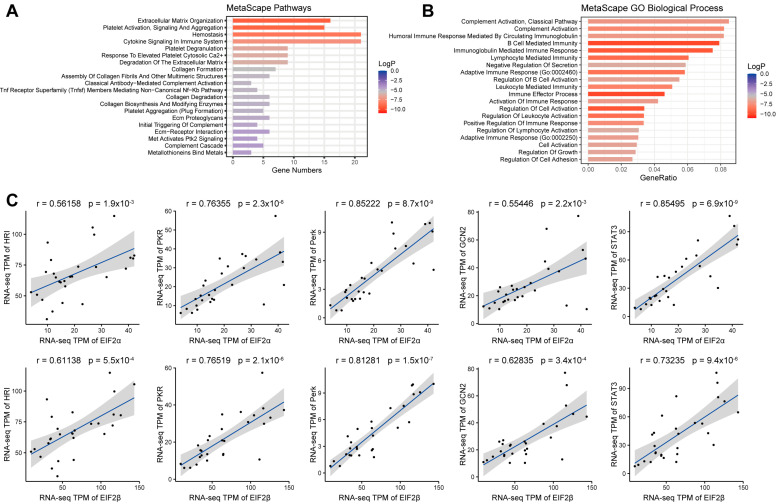
Differential gene-enrichment analyses of different EIF2β groups in 28 patients with somatotroph adenomas. (**A**,**B**): Metascape enrichment analysis, including pathways and GO terms. (**C**): Correlation analysis of EIF2α/β and EIF2AK1-4 and STAT3.

**Figure 4 ijms-23-13067-f004:**
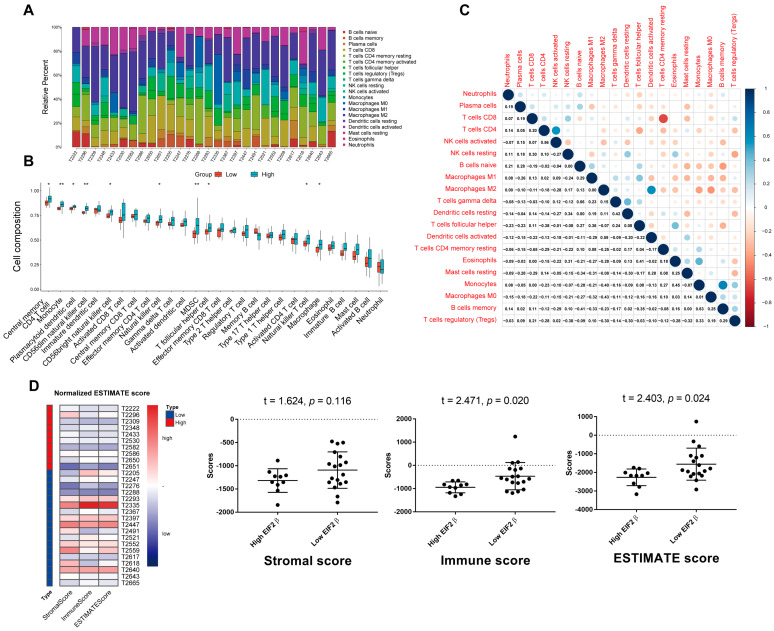
The immune landscape in the high- and low-EIF2β patient groups. (**A**): Heatmap from immune-cell signatures in 28 patients. (**B**): All 28 subtypes of tumor-infiltrating immune cells in patients with a high EIF2β compared with patients with a low EIF2β. The following conventions for symbols indicate statistical significance: blank: *p* > 0.05; *: *p* < 0.05; **: *p* < 0.01. (**C**): Heatmap showing that the ratios of the different tumor infiltration immune cell subgroups were moderately to strongly correlated. (**D**): Immune scores of 28 patients with somatotroph adenomas according to ESTIMATE.

**Figure 5 ijms-23-13067-f005:**
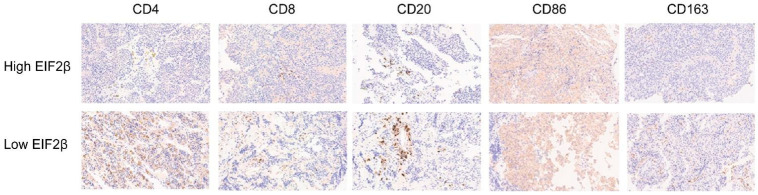
Immunohistochemistry images of CD4, CD8, CD20, CD86, and CD163 staining in the high- and low-EIF2β patient groups (×: 40 fold).

**Figure 6 ijms-23-13067-f006:**
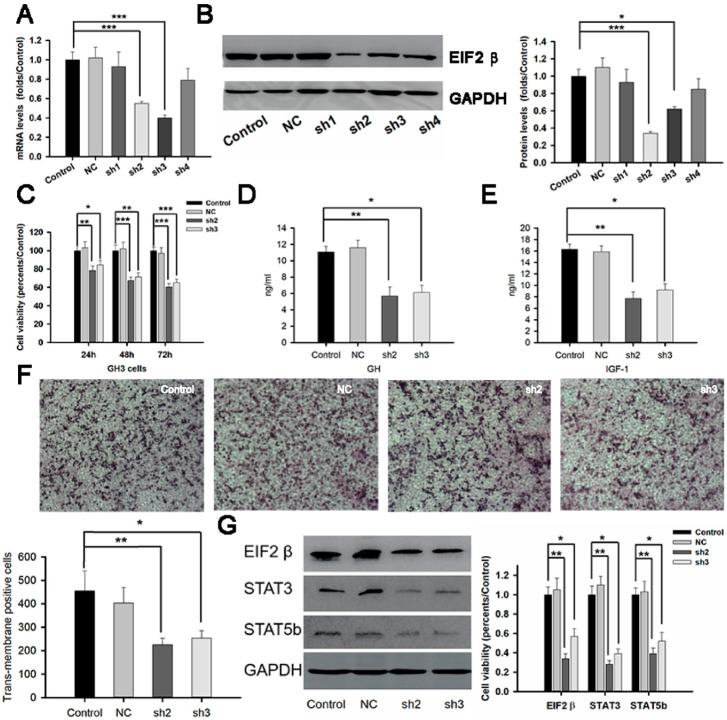
RNAi-EIF2β inhibited cell proliferation, growth hormone secretion, and invasion in vitro. (**A**,**B**): RT-PCR and Western blot experiments filtered the effective interference fragments of EIF2β in GH3 cells. (**C**): sh2-EIF2β and sh3-EIF2β reduced the cell viability of GH3 cells in a time-dependent manner. (**D**): sh2-EIF2β and sh3-EIF2β decreased growth hormone (GH) in GH3 cells. (**E**): sh2-EIF2β and sh3-EIF2β decreased insulin-like growth factor 1 (IGF-1) in GH3 cells. (**F**): sh2-EIF2β and sh3-EIF2β decreased trans-membrane-positive cells after 24 h treatment. (**G**): sh2-EIF2β and sh3-EIF2β reduced protein levels of STAT3 and STAT5b after 24 h treatment. * Compared to control group *p* < 0.05 ** *p* < 0.01 *** *p* < 0.001.

**Table 1 ijms-23-13067-t001:** Clinicopathological features of patients with pituitary adenomas.

Variable	CORT	GONA	SOMA	*p* Value
Sex				<0.001
Male	8	55	16	
Female	28	20	17	
Age	50.75 ± 2.10	52.85 ± 1.28	43.58 ± 1.89	<0.001
Tumor size				0.523
Micro	0	7	1	
Macro	29	66	29	
Giant	7	8	3	
Tumor volume (cm^3^)	14.39 ± 2.69	11.94 ± 2.62	6.89 ± 1.44	0.234
Invasion				0.004
No	11	46	20	
Yes	26	29	13	
Recurrence				0.022
No	23	65	26	
Yes	13	10	7	

Note: CORT: corticotroph adenoma; GONA: gonadotroph adenoma; SOMA: somatotroph adenoma.

**Table 2 ijms-23-13067-t002:** Clinically pathological features of patients with somatotroph adenoma.

Variable	EIF2β		*p* Value
	High	Low	
Sex			1
Male	5	8	
Female	5	8	
Age	42.4 ± 2.54	43.71 ± 2.87	0.76
Tumor size			0.274
Micro	0	1	
Macro	10	15	
Giant	0	3	
Tumor volume (cm^3^)	3.49 ± 0.66	7.76 ± 2.33	0.193
Invasion			0.038
No	3	14	
Yes	7	4	
Recurrence			1
No	7	14	
Yes	3	4	
Granulate			0.09
Sparseness	8	7	
Density	2	11	
HGH (folds/normal)	4.38 ± 1	2.32 ± 0.60	0.071

## Data Availability

All the data generated or analyzed in this study are included in this published article and its additional files.

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
