# Peer review of "The Integrated Stress Response Is Tumorigenic and Constitutes a Therapeutic Liability in Somatotroph Adenomas"

_ijms, 2022, doi:10.3390/ijms232113067_

Round 1

Reviewer 1 Report

This is a manuscript entitled “integrated stress response is tumorigenic and constitutes a therapeutic liability in somatotroph adenomas, by Zhenye Li et al.

MAIN PROBLEMS:

This is a study, demonstrating that the ISR is pivotal in somatotroph adenomas, and provide a rationale for the implementation of ISR-based regimens in future treatment strategies. However, in Lines 144-145 the authors state: “There was a tendency toward more sparsely granulated adenoma phenotype in 144 patients with high EIF2β than patients with low EIF2β.”.

Further (Lines 144-146), the state: “Patients with high EIF2β had more invasive potential (P = 0.038). There was a tendency toward more sparsely granulated adenoma phenotype in patients with high EIF2β than patients with low EIF2β. Therefore, the previous statement remains as speculation”. They state “..more sparsely granulated adenoma..”, but they do not explain from what other adenoma type this correlation refers.

ADDITIONAL WEAKNESS

Lines 10,11:

… “the nearly sparsely granulated somatotroph subtype being refractory to medication”. This is not true. According to the WHO calcification of pituitary adenomas of 2017, the sparsely granulated somatotroph subtype belong to High risk adenomas and they are less responsive to medical treatment.

For further details the authors can read the recent paper: PMID: 35061210

Please, revise.

Lines 14,15:

“differentiated  normal pituitary”.

By definition normal pituitary is composed of differentiated cells. Delete the word “differentiated ”

Lines 14,15:

Explain EIF2β by giving full name: Eukaryotic translation initiation factor 2β, and then put the abbreviation (EIF2β) in parenthesis.

Do exactly the same in the Introduction.

Line 21:

First explain and then abbreviate in parenthesis (HRI)

Do exactly the same in the Introduction.

Line 22:

First explain and then abbreviate in parenthesis (PKR)

Do exactly the same in the Introduction. None all readers are familiar with these abbreviations.

Line 75

The authors jump to Results without including a section MATERIAL AND METHODS.

What were the criteria of selection of the study group? In addition, they should explain in detail the method to diagnose and classify by histology and immunohistochemistry the pituitary adenoma types.

Lines 77-78:

They state: “Our study included 144 patients: 33 somatotroph adenomas, gonadotroph adenomas, and 36 corticotroph adenomas”. What about lactotroph adenomas (prolactinomas)? No one was included in their cohort?. Please explain and provide information regarding the criteria of selection of the study material. The absence of a pathologist among the authors looks strange.

Line 10:

The authors mention: sparsely granulated somatotroph adenomas”. What was the method used to difference densely from sparsely granulated somatotroph adenomas in their material?

Author Response

Comments and Suggestions for Authors

This is a manuscript entitled “integrated stress response is tumorigenic and constitutes a therapeutic liability in somatotroph adenomas, by Zhenye Li et al.

MAIN PROBLEMS:

This is a study, demonstrating that the ISR is pivotal in somatotroph adenomas, and provide a rationale for the implementation of ISR-based regimens in future treatment strategies. However, in Lines 144-145 the authors state: “There was a tendency toward more sparsely granulated adenoma phenotype in 144 patients with high EIF2β than patients with low EIF2β.”.

Further (Lines 144-146), the state: “Patients with high EIF2β had more invasive potential (P = 0.038). There was a tendency toward more sparsely granulated adenoma phenotype in patients with high EIF2β than patients with low EIF2β. Therefore, the previous statement remains as speculation”. They state “..more sparsely granulated adenoma..”, but they do not explain from what other adenoma type this correlation refers.

 Response:

Thank for the suggestion. According to Figure 3 and Table2, we focused the relationship of levels of eif2β and somatotroph adenoma phenotype. To avoid the mistaken, we have corrected the sentence.

The WHO 2017 classification suggested that subtypes of pituitary neuroendocrine tumors, such as male lactotroph, silent corticotroph and Crooke cell, sparsely granulated somatotroph, and silent plurihormonal PIT1 positive tumors, should be considered as "high risk" tumors. Based on the cognition, we think it was more important to focus the functional status (functional/silent) of corticotroph adenoma than granular kinds.

ADDITIONAL WEAKNESS

Question: Lines 10,11:… “the nearly sparsely granulated somatotroph subtype being refractory to medication”. This is not true. According to the WHO calcification of pituitary adenomas of 2017, the sparsely granulated somatotroph subtype belong to High risk adenomas and they are less responsive to medical treatment.

For further details the authors can read the recent paper: PMID: 35061210

Please, revise.

 Response:

Thank for the suggestion. We have re-edited the manuscript according to the suggestion.

Question: Lines 14,15:“differentiated  normal pituitary”.

By definition normal pituitary is composed of differentiated cells. Delete the word “differentiated ”

 Response:

Thank for the suggestion. We have re-edited the manuscript according to the suggestion.

Question: Lines 14,15:Explain EIF2β by giving full name: Eukaryotic translation initiation factor 2β, and then put the abbreviation (EIF2β) in parenthesis.

Do exactly the same in the Introduction.

 Response:

Thank for the suggestion. We have re-edited the manuscript according to the suggestion.

Question: Line 21:First explain and then abbreviate in parenthesis (HRI)

Do exactly the same in the Introduction.

 Response:

Thank for the suggestion. We have re-edited the manuscript according to the suggestion.

Question: Line 22:First explain and then abbreviate in parenthesis (PKR)

Do exactly the same in the Introduction. None all readers are familiar with these abbreviations.

 Response:

Thank for the suggestion. We have re-edited the manuscript according to the suggestion.

Question: Line 75:The authors jump to Results without including a section MATERIAL AND METHODS.

What were the criteria of selection of the study group? In addition, they should explain in detail the method to diagnose and classify by histology and immunohistochemistry the pituitary adenoma types.

 Response:

Thank for the suggestion. The structure of manuscript was: Introduction, Results, Discussion, Material and methods, reference according to the journal.

The pituitary adenomas types were diagnosed according to the 2017 World Health Organization classification of tumors of endocrine organs, including pituitary transcription factors and hormone content by immunohistochemistry staining.

We have added the sentence into the revised manuscript according the suggestion of reviewer.

Question: Lines 77-78:They state: “Our study included 144 patients: 33 somatotroph adenomas, gonadotroph adenomas, and 36 corticotroph adenomas”. What about lactotroph adenomas (prolactinomas)? No one was included in their cohort?. Please explain and provide information regarding the criteria of selection of the study material. The absence of a pathologist among the authors looks strange.

 Response:

Thank for the suggestion.

In this study, omics data mainly included transcriptomics and mass spectrometry. In the part of mass spectrometry, we only incorporated somatotroph, gonadotroph and corticotroph adenoma into the cohort, and excluded the lactotroph adenomas based on too small the tumor size. In fact, this was our pity.

We co-signed with the pathologist in many papers. In this manuscript, we thanked the pathologists, Professor Guilin Li and Yanjiao He in Acknowledgments.

Question: Line 10:The authors mention: sparsely granulated somatotroph adenomas”. What was the method used to difference densely from sparsely granulated somatotroph adenomas in their material?

 Response:

Thank for the suggestion. Somatotroph adenomas are divided into two distinct and clinically relevant histologic subtypes, densely and sparsely granulated, based on the density of GH-containing secretory granules and the presence of fibrous bodies.

We have added the sentence into the revised manuscript according to the suggestion of reviewer.

Reviewer 2 Report

Review

The integrated stress response is tumorigenic and constitutes a therapeutic liability in somatotroph adenomas

The authors in this paper tried to explore the role of EIF2, a factor involved in the integrated stress response pathway (ISR) in somatotroph adenomas, by using various “multi-omics” approach. While the language and methodologies in the paper are sound, some inconsistencies are noted, as written below.

  1. Lines 13 - 15: “we identified the characteristic profiling of ribosome structural dynamics,” no such study is found in this paper. In addition, although it was mentioned that other adenomas were used to compare with the normal pituitary using proteomics, only the results of somatotroph adenomas were shown.

  2. Lines 20- 21: “showed uncovered” - synonyms.

  3. Expand the abbreviations PERK, GCN2, PKR, and HRI during the initial introduction.

  4. Table 1 shows a significant difference in sex and age parameters. This was not explained.

  5. What are the nanoLC-MS/MS results for gonadotroph adenomas and five corticotroph samples?

  6. Line 106: Correct “green” to “blue.”

  7. State the reason what finding in section 2.2 prompted to study EIF2β specifically.

  8. Why weren't EIF2α and p-EIF2α, crucial factors in the ISR pathway, studied? 

  9. Fig.2: A control/normal pituitary IHC image of EIF2β and STAT3 will be helpful. 

  10. Fig. S1: Show the linear correlation. 

  11. What are the cut-off points/values for the high and low-level expression of EIF2β?

  12. Fig. 4B: x-axis values are not visible. 

  13. What is the possible explanation that STAT3 and STAT5B were reduced with EIF2β?

  14. Fig. 6F: Please add the methodology for measuring trans-membrane-positive cells. Also, label the corresponding times on the images.

  15. Line 225: There were no in-vivo experiments in this manuscript. You can modify it to ex-vivo.

  16. Lines 236 - 237: Please refer to the experiment where they were studied.

Author Response

Question: Lines 13 - 15: “we identified the characteristic profiling of ribosome structural dynamics,” no such study is found in this paper. In addition, although it was mentioned that other adenomas were used to compare with the normal pituitary using proteomics, only the results of somatotroph adenomas were shown.

 Response:

Thank for the suggestion. We are sorry for the writing error, and we re-edited the sentence as follows: In this study, we identified the characteristic profiling of integrated stress response in translocation and translation initiation factor activity........

In this study, the differentially expressed protein refers to the somatotroph adenoma compared to gonadotroph, corticotroph and normal pituitary gland, We have corrected the writing error in Figure 1B.

Question: Lines 20- 21: “showed uncovered” - synonyms.

 Response:

Thank for the suggestion.We have deleted the redundant word according to the suggestion of reviewer.

Question:Expand the abbreviations PERK, GCN2, PKR, and HRI during the initial introduction.

 Response:

Thank for the suggestion. We have re-edited the manuscript according to the suggestion of reviewer.

Question:Table 1 shows a significant difference in sex and age parameters. This was not explained.

 Response:

Thank for the suggestion. We have added the result into the revised manuscript as follows: There were more female patients and more invasion and recurrence chance in corticotroph adenoma compared, and more younger patient in somatotroph adenoma (P<0.05).

Question:What are the nanoLC-MS/MS results for gonadotroph adenomas and five corticotroph samples?

 Response:

Thank for the suggestion. In this study, the differentially expressed protein refers to the somatotroph adenoma compared to gonadotroph, corticotroph and normal pituitary gland, We have corrected the writing error in Figure 1B.

Question:Line 106: Correct “green” to “blue.”

 Response:

Thank for the suggestion. We have corrected the writing error according to the suggestion of reviewer.

Question:State the reason what finding in section 2.2 prompted to study EIF2β specifically.

 Response:

Thank for the suggestion. Mass spectrometric data showed EI2β was one of differentially expressed proteins, not EI2α or EI2γ.

We have added the sentence into the revised manuscript according to the suggestion of reviewer.

Question:Why weren't EIF2α and p-EIF2α, crucial factors in the ISR pathway, studied? 

 Response:

Thank for the suggestion.  Mass spectrometric data showed EIF2 is a key component of the ternary complex whose role is to deliver initiator tRNA into the ribosome. Mass spectrometric data showed increased subunit α/β and reduction in γ subunit, mainly subunit . In addition, correlation analysis of transcriptome data also showd the priority of EIF2β compared with EIF2α in Figure 3C.

Furthermore, we did not detect the difference of  p-EIF2α base on our previous phosphorylation mass spectrometry.

Question:Fig.2: A control/normal pituitary IHC image of EIF2β and STAT3 will be helpful. 

 Response:

Thank for the suggestion. We also are sorry for the lack of normal pituitary gland, At present, we try to find normal pituitary specimens based on the body donation program.

Question: Fig. S1: Show the linear correlation. 

 Response:

Thank for the suggestion. We have added the linear correlation in the revised manuscript according to the suggestion of reviewer.

Question:What are the cut-off points/values for the high and low-level expression of EIF2β?

 Response:

Thank for the suggestion. The cut-off value was (H-score 114), and We have added it in the revised manuscript according to the suggestion of reviewer.

Question:Fig. 4B: x-axis values are not visible. 

 Response:

Thank for the suggestion. We have enlarged the font size of Figure 4B according to the suggestion of reviewer.

Question:What is the possible explanation that STAT3 and STAT5B were reduced with EIF2β?

 Response:

Thank for the suggestion. We guessed EIF2β was mainly involved in tumor-related pathways in somatotroph adenomas, including the cell adhesion molecule pathway and the JAK-STAT signaling pathway. We speculated that eIF2β might inhibit T cell-mediated immunity by promoting the Treg response. In addition, the expression of eIF2β was positively correlated with STAT3 and STAT5b. Related studies have shown that some signaling molecules are involved in M2 polarization of macrophages, such as PI3K/AKT-ERK signaling, STAT3, HIF1α, and STAT6. These results suggested that eIF2β might regulate tumor macrophage infiltration, which would have effects on the tumor microenvironment.

Question:Fig. 6F: Please add the methodology for measuring trans-membrane-positive cells. Also, label the corresponding times on the images.

 Response:

Thank for the suggestion. We have added he methodology and re-edited Figure 6 according to the suggestion of reviewer.

 Question:Line 225: There were no in-vivo experiments in this manuscript. You can modify it to ex-vivo.

 Response:

Thank for the suggestion. We have corrected the writing error according to the suggestion of reviewer.

Question:Lines 236 - 237: Please refer to the experiment where they were studied.

 Response:

Thank for the suggestion. We have added “ as shown in Figure 1C” according to the suggestion of reviewer.

Round 2

Reviewer 1 Report

The term high risk was included only in the Abstract (sparsely granulated somatotroph subtype belong to High risk adenomas and they are less responsive to medical treatment).

The authors should include all categories of high risk adenomas in the manuscript and quote the indicated reference (PMID: 35061210) in the text and in the Reference list.

The WHO 2017 classification suggested that subtypes of pituitary neuroendocrine tumors, such as male lactotroph, silent corticotroph and Crooke cell, sparsely granulated somatotroph, and silent plurihormonal PIT1 positive tumors, should be considered as "high risk" tumors. 

The term pituitary neuroendocrine tumors is not included in the WHO 2017 classification. Replace it by adenomas.

The authors have "excluded the lactotroph adenomas based on too small the tumor size". This is partly true. Lactotroph adenomas nearly always present as large tumors in males. In contrast, as a rule corticotrroph adenomas are microadenomas in their substantial majority. Therefore, they must include lactotroph macrioadenomas in their study..

Author Response

Question 1: The term high risk was included only in the Abstract (sparsely granulated somatotroph subtype belong to High risk adenomas and they are less responsive to medical treatment).

Response: Thank you for the suggestion. We have added the detailed description about the sparsely granulated somatotroph subtype into the manuscript according to the suggestion.

Question 2: The authors should include all categories of high risk adenomas in the manuscript and quote the indicated reference (PMID: 35061210) in the text and in the Reference list.

Response: Thank you for the suggestion. We have added the description and cited the paper (PMID: 35061210) into the manuscript according to the suggestion.

Question 3: The WHO 2017 classification suggested that subtypes of pituitary neuroendocrine tumors, such as male lactotroph, silent corticotroph and Crooke cell, sparsely granulated somatotroph, and silent plurihormonal PIT1 positive tumors, should be considered as "high risk" tumors.

Response: Thank you for the suggestion. We have added the description of "high risk" tumors into the manuscript according to your helpful suggestion.

Question 4: The term pituitary neuroendocrine tumors is not included in the WHO 2017 classification. Replace it by adenomas.

Response: Thank you for the suggestion. We have corrected the error according to the suggestion.

Question 5: The authors have "excluded the lactotroph adenomas based on too small the tumor size". This is partly true. Lactotroph adenomas nearly always present as large tumors in males. In contrast, as a rule corticotrroph adenomas are microadenomas in their substantial majority. Therefore, they must include lactotroph macrioadenomas in their study.

Response: Thank you for the suggestion. We were not able to verify this in nanoLC-MS because of the amount of lactotroph adenomas tissues. We have supplemented the transcriptome data including the male lactotroph into the manuscript according to the suggestion as S1.

Reviewer 2 Report

The integrated stress response is tumorigenic and constitutes a therapeutic liability in somatotroph adenomas. 

Although the authors were able to address the concerns that were raised in the original manuscript, there needs to be the inclusion of a proper explanation accordingly. Below are my comments:

  • Please get the manuscript verified by a native English writer.

  • Clearly explain the sex, age, and recurrence significance. Introduce the statistical significance observed for age, sex, and recurrence, as evidenced in the table, and then specify each category. 

  • You mentioned you collected five normal pituitary glands. Was IHC not performed on them?

  • Mention that the cut-off H-score is 114; a score above 114 is high, and a below is low

  • Please rewrite the cell migration methodology and use scientifically appropriate terms (“trans-membrane positive cells” might be the correct term)

  • Label the control, NC, and KD, not the 24 hr time in 6F

  • “Transwell experiments showed that the trans-membrane cells in the RNAi-EIF2β groups were reduced compared with the control group (Figure 6F).”  - Rewrite in a scientifically accurate way.

  • Complete the proteomics analysis methodology, virus packaging, and ELISA.

  • You mentioned the reason for the decreased expression of STAT3 and STAT5b; please add the explanation in the manuscript.

Author Response

Question 1: Please get the manuscript verified by a native English writer.

Response: Thank you for the suggestion. We have sent the article to be polished by the English language editing company as requested by the magazine.

Question 2: Clearly explain the sex, age, and recurrence significance. Introduce the statistical significance observed for age, sex, and recurrence, as evidenced in the table, and then specify each category. 

Response: Thank you for the suggestion. We have added the detailed description about the table 1 into the manuscript according to the suggestion.

Question 3: You mentioned you collected five normal pituitary glands. Was IHC not performed on them?

Response: Thank you for the question. Our normal pituitary tissues were not the whole pituitary gland tissue, and the amount of each sample was limiting. We barely have enough tissue to do IHC after mass spectrometry.

Question 4: Mention that the cut-off H-score is 114; a score above 114 is high, and a below is low

Response: Thank you for the question. 114 was the median of H-score, so we were divided into high and low groups according to this value.

Question 5: Please rewrite the cell migration methodology and use scientifically appropriate terms (“trans-membrane positive cells” might be the correct term)

Response: Thank you for the suggestion. We have rewrite the cell migration methodology and use the “trans-membrane positive cells” correct term into the manuscript according to the suggestion.

Question 6: Label the control, NC, and KD, not the 24 hr time in 6F

Response: Thank you for the suggestion. We have corrected the error of Figure 6F according to the suggestion.

Question 7: “Transwell experiments showed that the trans-membrane cells in the RNAi-EIF2β groups were reduced compared with the control group (Figure 6F).”  - Rewrite in a scientifically accurate way.

Response: Thank you for the suggestion. We have added a statistical description into the manuscript according to the suggestion.

Question 8: Complete the proteomics analysis methodology, virus packaging, and ELISA.

Response: Thank you for the suggestion. We have added detailed methodological description into the manuscript according to the suggestion.

Question 9: You mentioned the reason for the decreased expression of STAT3 and STAT5b; please add the explanation in the manuscript.

Response: Thank you for the suggestion. We have added the decreased expression of STAT3 and STAT5b we mentioned in the last reply into the manuscript according to the suggestion.

Round 3

Reviewer 1 Report

The authors have adopted the comments and made the appropriate additions-changes in the manuscript.